# Assessment of Sarcopenia in the Intensive Care Unit and 1-Year Mortality in Survivors of Critical Illness

**DOI:** 10.3390/nu13082726

**Published:** 2021-08-08

**Authors:** Naoya Yanagi, Tomotaka Koike, Kentaro Kamiya, Nobuaki Hamazaki, Kohei Nozaki, Takafumi Ichikawa, Atsuhiko Matsunaga, Masayuki Kuroiwa, Masayasu Arai

**Affiliations:** 1Department of Rehabilitation Sciences, Graduate School of Medical Sciences, Kitasato University, Sagamihara 252-0373, Japan; ap15335@st.kitasato-u.ac.jp; 2Department of Intensive Care Center, Kitasato University Hospital, Sagamihara 252-0329, Japan; tkoike@kitasato-u.ac.jp; 3Department of Rehabilitation, School of Allied Health Sciences, Kitasato University, Sagamihara 252-0373, Japan; atsuhikonet@gmail.com; 4Department of Rehabilitation, Kitasato University Hospital, Sagamihara 252-0329, Japan; hamanobu0317@gmail.com (N.H.); 0818.n.kohei@gmail.com (K.N.); takafumi@kitasato-u.ac.jp (T.I.); 5Department of Anesthesiology, School of Medicine, Kitasato University, Sagamihara 252-0373, Japan; masa9618@yahoo.co.jp; 6Division of Intensive Care Medicine, Department of Research and Development Center for New Medical Frontiers, School of Medicine, Kitasato University, Sagamihara 252-0373, Japan; hiroma557@yahoo.co.jp

**Keywords:** critical illness, sarcopenia, skeletal muscle mass, skeletal muscle function, ultrasound

## Abstract

Skeletal muscle wasting in the intensive care unit (ICU) has been associated with mortality, but it is unclear whether sarcopenia, defined by skeletal muscle mass and function, is useful for detailed risk stratification after ICU discharge. In this cohort study, 72 critically ill patients with an ICU stay of ≥48 h were identified. Skeletal muscle mass was assessed from the muscle thickness (MT) of the patients’ quadriceps using ultrasound images before ICU discharge. Skeletal muscle function was assessed from the patients’ muscle strength (MS) before ICU discharge according to the Medical Research Council sum score. A diagnosis of sarcopenia in the ICU was made in patients with low MT and low MS. The study endpoint was 1-year mortality. Sarcopenia in the ICU was diagnosed in 26/72 patients (36%). After adjusting for covariates in the Cox regression, sarcopenia in the ICU was significantly associated with 1-year mortality (hazard ratio 3.82; 95% confidence interval, 1.40–10.42). Sarcopenia in the ICU, defined by low skeletal muscle mass and function, was associated with 1-year mortality in survivors of critical illness. Skeletal muscle mass and function assessed at the bedside could be used to identify higher-risk patients in the ICU.

## 1. Introduction

Skeletal muscle wasting and the development of intensive care unit-acquired weakness (ICU-AW) dramatically affect mortality and mobility disability after ICU discharge [1,2,3]. Moreover, skeletal muscle mass at ICU admission has been associated with mortality [4,5,6]. Thus, the value of assessing skeletal muscle mass and function in ICU patients has been emphasized in recent years [7].

Sarcopenia is defined as the loss of skeletal muscle mass and function [8]. It is described as a predictive biomarker that can help identify the risk of mobility disability and prognosis in hospitalized patients [9]. Several previous studies have assessed sarcopenia in ICU patients and have shown that it is associated with a poor prognosis [10,11,12,13]. However, sarcopenia was defined by skeletal muscle mass alone, or laboratory data from previous studies among patients with critical illnesses.

We hypothesized that both the assessment of skeletal muscle mass and function, according to the definition of sarcopenia, would provide detailed risk stratification in ICU patients. However, there are no reports of the association between sarcopenia in the ICU and mortality, as defined by the assessment of both skeletal muscle mass and function. The aim of the study was to investigate whether the available assessment of sarcopenia in the ICU could help identify patients at high risk of 1-year mortality among survivors of critical illness.

## 2. Materials and Methods

### 2.1. Study Design, Setting and Populations

This single-center observational study examined a cohort of consecutive patients who were admitted to the surgical and medical ICUs of Kitasato University Hospital between August 2017 and May 2019 and underwent ultrasound measurements within 1 week of ICU admission. Ultrasound measurements were performed on patients who were predicted to stay ≥48 h in the ICU. Patients who could not independently perform basic activities of daily living (ADLs), including walking with assistance before ICU admission and those who died in the ICU were excluded. This study followed the Strengthening Reporting of Observational Studies in Epidemiology (STROBE) reporting guidelines for cohort studies [14]. The study protocol conformed to the Declaration of Helsinki and was approved by the Ethics Committee of Kitasato University Hospital (B18–162).

### 2.2. Data Collection

Data were obtained on all baseline information, including demographic characteristics such as age, sex, body mass index (BMI), ADLs, and modified Rankin Scale score [15] prior to hospitalization from the electronic medical records. In addition, critical illness data on the reason for ICU admission, Acute Physiology and Chronic Health Evaluation (APACHE) II score [16] at ICU admission, maximal sequential organ failure assessment (SOFA) score [17] during the ICU stay, length of the ICU stay, the requirement of extracorporeal membrane oxygenation (ECMO), and use of noradrenaline during the ICU stay were collected from the electronic medical records.

### 2.3. Ultrasound Measurement

Skeletal muscle mass was assessed from the muscle thickness (MT) of the patient’s quadriceps assessed using B-mode ultrasound images before ICU discharge. We defined the patients’ MT as the total thickness of the rectus femoris and vastus intermedius. Ultrasound (Vscan ^®^Dual Probe) of 3.4–8.0 MHz was used to obtain images. All images were measured by an experienced assessor using the same settings. Patients were supine with their knees extended, and the assessor identified the measurement point as the midpoint between the superior anterior iliac spine and superior border of patella. The probe was placed vertically to the femur, and transverse images of the MT were obtained. A large volume of gel was used to minimize the pressure on the tissue. The validity of the ultrasound method to assess skeletal muscle in critically ill patients was assessed by two physical therapists, who analyzed the patients’ MT images to evaluate the internal validity of the measurements. As a result, high internal validity for analyzing the obtained images was achieved, with an intraclass correlation coefficient of 0.951.

### 2.4. Definition of Sarcopenia in the ICU

Skeletal muscle function was identified from the patients’ muscle strength (MS) before ICU discharge, according to manual muscle testing done based on the Medical Research Council (MRC) sum score [18]. The MRC sum score assesses the muscle strength of each muscle group in the upper and lower limbs with total scores ranging from 0 (worst) to 60 (best). Scores for each muscle group ranged from 0 to 5, with higher scores indicating greater muscle strength. When assessing MS, based on MRC sum score, the patients were also assessed for their ability to focus attention on verbal commands according to the Richmond Agitation Sedation Scale (RASS) [19]. If the patients’ RASS score ranged from −1 to +1, MRC score was measured.

Sarcopenia in the ICU was defined using the assessments of MT and MS. MT was categorized as low or high based on the median for each sex, and MS was categorized as low based on an MRC sum score of <48 [20]. A diagnosis of sarcopenia in the ICU was made in patients with low MT and low MS. In addition, we categorized the patients into four groups according to MT and MS.

### 2.5. Endpoints

The study endpoint was all-cause mortality, which was identified through a medical chart review. The duration of these events was calculated as the number of days from ICU discharge to the event’s date. The follow-up period was set at 1 year.

### 2.6. Statistical Methods

Continuous variables are expressed as median with interquartile range (IQR), and categorical variables are expressed as *n* (%). Continuous variables were compared between the four groups using one-way analysis of variance and the Kruskal–Wallis test as appropriate, while the χ^2^ test was used for categorical variables. The association between sarcopenia in the ICU and 1-year mortality, the Kaplan–Meier method with the log-rank test and Cox proportional hazard model was used. To avoid overfitting, the propensity score estimated from all potential confounding risk factors (age, sex, BMI, APACHE II score, and SOFA score) was adjusted [21]. In addition, receiver operating characteristic (ROC) curves for 1-year mortality were constructed to determine whether MS complemented the predictive abilities of MT. The areas under the ROC curves (AUCs) were compared according to the DeLong method [22]. Furthermore, we calculated the incremental information of adding MS to MT with the use of net reclassification improvement (NRI) and integrated discrimination improvement (IDI), which were used to compare the accuracy of the models in more detail [23]. Statistical analyses were performed using Stata version 16.1 (Stata Corp, College Station, TX, USA). A two-tailed *p* value < 0.05 was considered statistically significant in all analyses.

## 3. Results

### 3.1. Patient Characteristics

In this study, 72 patients met the inclusion criteria. The median age of the study population was 70 years (IQR: 60–76), the median APACHE II score at admission was 23 (IQR: 18–28), the maximum SOFA score was 11 (IQR: 8–14), and 57/72 (79%) were male. The reasons for ICU admission were as follows: sepsis, 23/72 (32%); following cardiac surgery, 23/72 (32%); respiratory failure, 20/72 (28%); and other reasons, 6/72 (8%). The median ICU length of stay was 8 days (IQR: 4–12) (Table 1). Table 2 presents the comparison of patient characteristics among the four groups. A diagnosis of sarcopenia in the ICU was made in 26/72 patients (36%).

### 3.2. Sarcopenia in the ICU and 1-Year Mortality

Over a mean follow-up period of 180 ± 124 days, 19 patients died. Sarcopenia, defined by low MT and low MS in the ICU, was significantly associated with 1-year mortality on the Kaplan–Meier curves and log-rank tests (*p* < 0.001; Figure 1). The prognosis was also poor in the low MS-only and low MT-only groups, but sarcopenic patients who had both factors together had an especially poor prognosis. After adjusting for age, sex, BMI, APACHE II score at admission, and maximum SOFA score, sarcopenia in the ICU was significantly associated with 1-year mortality (vs. non-sarcopenia: hazard ratio [HR] 3.82; 95% confidence interval [CI], 1.40–10.42; *p* = 0.009).

### 3.3. Additive Prognostic Predictive Capabilities of Sarcopenia Assessment in the ICU

The AUC was greatest with the addition of MS to MT, although the difference was not statistically significant (MT only: AUC 0.73, 95% CI 0.57–0.88; MT + MS: AUC 0.84, 95% CI 0.73–0.94; *p* = 0.132; Figure 2). In detailed analyses, the prognostic predictive capability was significantly improved in the model of MT + MS in both NRI and IDI for 1-year mortality (NRI 0.95, 95% CI 0.69–1.22, *p* < 0.001; IDI 0.23, 95% CI 0.16–0.29, *p* < 0.001).

## 4. Discussion

This study showed that the assessment of sarcopenia defined by low skeletal muscle mass and function in the ICU was useful for risk stratification in critical illness survivors. The combination of skeletal muscle mass and function assessment improved the prognostic predictive capability for 1-year mortality in critical illness survivors.

Many previous studies have documented an association between muscle wasting and mortality in patients with critical illnesses, which were consistent with our findings [10,11,12,23,24,25]. A prospective cohort study showed that the rectus femoris cross-sectional area measured using ultrasound predicted adverse discharge disposition in surgical ICU patients [10]. Another study reported that the psoas muscle areas assessed using computed tomography (CT) had predictive value for weaning outcomes from mechanical ventilation and mortality [25]. In addition, ICU-AW, defined by systemic muscle weakness, has been widely reported to be associated with prognosis [1,2,3]. However, the assessment of sarcopenia in the ICU has been defined by skeletal muscle mass or muscle weakness alone in previous studies.

Sarcopenia is registered as a disease in the ICD-10 code, which has recently been reported in community-dwelling older people and hospitalized patients, and the standard definition is by both a measure of skeletal muscle mass and function. To our knowledge, few studies have assessed the loss of skeletal muscle mass and function in patients with critical illness according to the standard concept of sarcopenia and reported an association between sarcopenia and prognosis. In the results of the present study, low skeletal muscle mass or muscle dysfunction alone had a poor prognosis, and this prognosis was even worse in patients with both. Therefore, the assessment of sarcopenia, according to the standard definition as both a measure of skeletal muscle mass and function adopted in this study may be one of the recommendations in the ICU setting.

In this study, patients with sarcopenia in the ICU were older, had a higher SOFA score, and a higher percentage were treated with noradrenaline. This suggests that patients with more severe illness are more likely to develop a low classification for both skeletal muscle mass and function. Although the small sample size of this study limits the investigation of the association, previous studies [26,27] have shown that multiorgan system failure and septic shock may be triggers for the development of sarcopenia in the ICU. The findings of this study suggest that patients with more severe illnesses should be managed for sarcopenia in the ICU. There is insufficient evidence for the management of sarcopenia in the ICU. Thus, further research is needed to determine whether various treatments, such as early rehabilitation, neuromuscular electrical stimulation, nutritional therapy, and glycemic control, are effective in preventing and treating sarcopenia.

A recent study of 248 adult patients with critical illnesses showed that physical disability, defined by low gait speed, was associated with mortality among post-intensive care syndrome components [28]. ICU-AW and patients with low skeletal muscle mass frequently develop physical disability after ICU discharge [2,29,30]. Thus, in this study, sarcopenia in the ICU may be associated with mortality due to physical disability after ICU discharge.

This study had several limitations. First, this was a single-center observational study, and the design precluded the determination of a causal association between sarcopenia in the ICU and 1-year mortality. Second, the study population was smaller than that in previous studies [6,10,11]. Based on the results obtained, we believe it is necessary to replicate the study on larger sample sizes in the future. However, patients with a high severity of illness were included in this study compared to previous studies. The study findings may apply to ICU patients with higher illness severity. Third, this study could not assess the low physical function required for the definition of sarcopenia. In future studies, it is suggested to conduct a more detailed assessment of sarcopenia through an evaluation of the low physical function in the ICU. Finally, the study’s available data may not have been fully adjusted for factors associated with mortality. To examine the findings of this study, the presence of some unmeasured confounding factors (e.g., education level, frailty status, or sarcopenia before ICU admission) must be considered. Further research on frailty or sarcopenia measures prior to ICU admission is required.

## 5. Conclusions

Sarcopenia in the ICU, defined by low skeletal muscle mass and function, was associated with 1-year mortality in critical illness survivors. The combination of skeletal muscle mass and function assessment significantly improved the accuracy of the predictive prognostic capability for 1-year mortality. These results indicate the importance of sarcopenia assessment based on skeletal muscle mass and function, which has been proposed for the elderly and other medical conditions, even in ICU patients.

## Figures and Tables

**Figure 1 nutrients-13-02726-f001:**
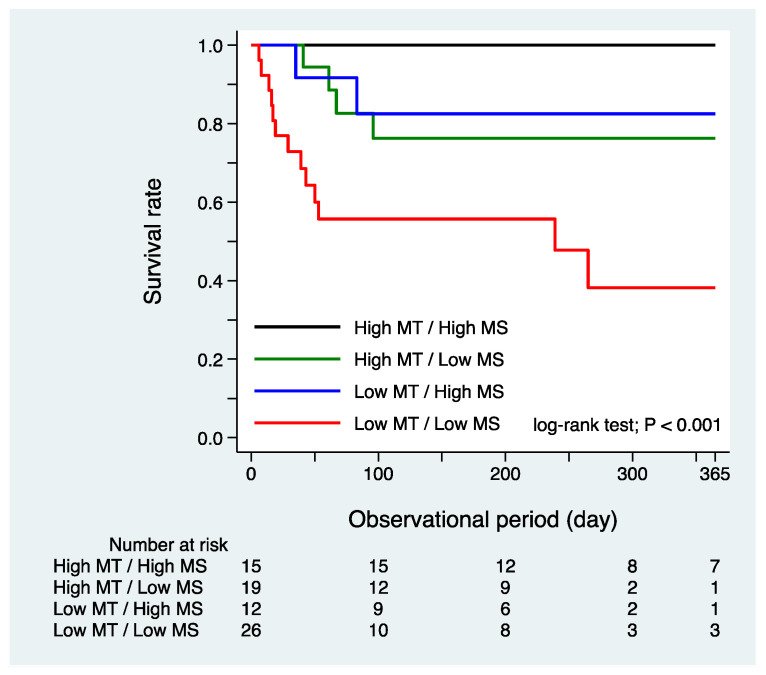
Kaplan–Meier analysis of 1-year mortality according to sarcopenia defined by low muscle mass and strength in the ICU. MT: muscle thickness, MS: Muscle strength.

**Figure 2 nutrients-13-02726-f002:**
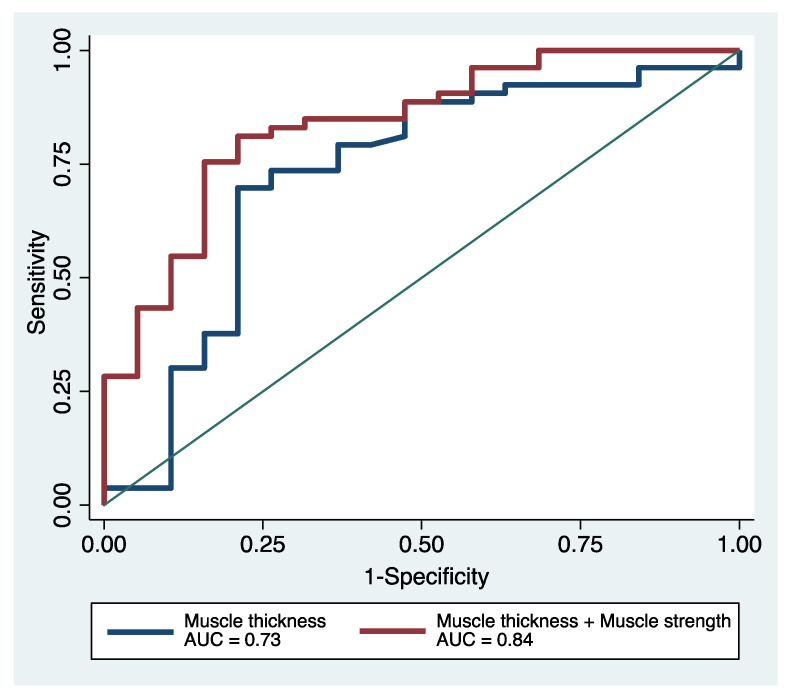
Receiver operating characteristics curves of muscle thickness only and muscle thickness plus muscle strength for 1-year mortality. AUC: area under the receiver operating characteristics curve.

**Table 1 nutrients-13-02726-t001:** Patient characteristics.

Characteristics	Overall (*n* = 72)	
Age (years), median [IQR]	70	(60–76)
Male, n (%)	57	(79)
BMI (kg/m^2^), median [IQR]	23.6	(20.8–26.6)
Reason for ICU admission, n (%)		
Sepsis	23	(32)
Cardiac surgery	23	(32)
Respiratory failure	20	(28)
Others	6	(8)
ICU LOS (days), median [IQR]	8	(4–12)
mRS prior to admission, median [IQR]	2	(1–2)
APACHE II score at admission, median [IQR]	23	(18–28)
Maximum SOFA score, median [IQR]	11	(8–14)
ECMO, n (%)	4	(6)
Noradrenaline, n (%)	33	(46)
MRC sum score, median [IQR]	42	(36–48)
Muscle thickness (mm), median [IQR]	20.9	(16.0–27.1)
Death, n (%)	19	(26)

IQR: interquartile range, BMI: body mass index, ICU: intensive care unit, LOS: length of stay, mRS: modified Rankin Scale, APACHE: Acute Physiology and Chronic Health Evaluation, SOFA: Sequential Organ Failure Assessment, ECMO: extracorporeal membrane oxygenation, MRC: Medical Research Council.

**Table 2 nutrients-13-02726-t002:** Comparison of patient characteristics between the four groups.

Characteristics	High MT and MS (*n* = 15)		High MT and Low MS (*n* = 19)		Low MT and High MS (*n* = 12)		Low MT and Low MS (Sarcopenia) (*n* = 26)		*p* Value
Age (years), median [IQR]	60	(51–74)	70	(53–73)	70	(62–73)	73	(69–77)	0.051
Male, n (%)	11	(73)	16	(84)	9	(75)	21	(81)	0.857
BMI (kg/m^2^), median [IQR]	25.1	(21.4–27.3)	26.2	(24.1–28.3)	20.2	(19.0–23.5)	22.8	(19.8–25.2)	0.004
Reason of ICU admission, n (%)									
Sepsis	1	(7)	6	(32)	5	(42)	11	(42)	0.006
Cardiac Surgery	7	(47)	8	(42)	1	(8)	6	(23)	0.220
Respiratory Failure	4	(27)	3	(16)	2	(17)	7	(27)	0.795
Others	3	(20)	2	(11)	4	(33)	2	(8)	0.072
ICU LOS (days), median [IQR]	4	(3–5)	11	(7–15)	5	(4–9)	10	(7–12)	0.014
mRS prior to admission, median [IQR]	1	(0–3)	1	(0–2)	2	(1–2)	3	(2–3)	0.588
APACHE II score at admission, median [IQR]	21	(10–26)	24	(9–13)	22	(19–26)	23	(19–26)	0.164
Maximum SOFA score, median [IQR]	8	(5–13)	11	(9–13)	11	(7–13)	12	(10–15)	0.087
ECMO, n (%)	0	(0)	1	(5)	1	(8)	2	(8)	0.732
Noradrenaline, n (%)	2	(13)	8	(42)	7	(58)	16	(62)	0.016
Death, n (%)	0	(0)	4	(21)	2	(17)	13	(50)	0.002

IQR = interquartile range, BMI = body mass index, ICU = intensive care unit, LOS = length of stay, mRS = modified Rankin Scale, APACHE = Acute Physiology and Chronic Health Evaluation, SOFA = Sequential Organ Failure Assessment, ECMO = extracorporeal membrane oxygenation, MT = Muscle thickness, MS = Muscle strength.

## Data Availability

Data cannot be shared publicly because of the restrictions by the Ethics Committee of Kitasato University Hospital.

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
