# Peer review of "Assessment of Sarcopenia in the Intensive Care Unit and 1-Year Mortality in Survivors of Critical Illness"

_nutrients, 2021, doi:10.3390/nu13082726_

Round 1

Reviewer 1 Report

In this observational study Yanagi et al. investigated whether the available assessment of sarcopenia in the ICU could help identify patients at high risk of 1-year mortality among survivors of critical illness.

Sarcopenia was assessed by combining the skeletal muscle thickness of the patients' quadriceps using ultrasound images, and skeletal muscle function according to the Medical Research Council sum score. Authors observed that the presence of sarcopenia in the ICU was significantly associated with 1-year mortality (HR: 3.82; CI: 1.40-10.42).

Major issues: the major concerns of this study is the very limited number of patients (72) and the methods used for the evaluation of sarcopenia

Minor issues:

  1. Abstract: Skeletal muscle mass was assessed... Before the discharge? Please add!
  2. Introduction: reference 8 should be updated with the consensus of 2019 (J Am Med Dir Assoc 2020 Mar;21(3):300-307.e2. doi: 10.1016/j.jamda.2019.12.012.), and sarcopenia should be defined more in detail as an age-related loss of muscle mass, plus low muscle strength, and/or low physical performance.
  3. Material and Method:
  • Pleased define the manual muscle testing.
  • I think that reference 18 is not correct.
  • The four groups reported in table 2 should be define in this section.
  • Was the sample size calculated?
  1. Discussion: authors stated “The findings of this study suggest that patients with more severe illness should be managed for sarcopenia in the ICU” (line 184). What could be the management of sarcopenia in the ICU, considering that, so far, only training exercise is considered the main approach to sarcopenia prevention/treatment?

Reviewer 2 Report

An interesting paper with preliminary findings that suggest a relationship between sarcopenia and one year survival post ICU discharge.

Major comments

One of the biggest limitations of this study is the sample size separated into 4 groups. As a result, the authors need to emphasize more that these results are preliminary and need to be confirmed with a larger study.

Exactly how was muscle strength determined? Details need to be provided as to the procedures as well as how low muscle strength was defined. Was there a cut-off for RASS score when to assess muscle strength? Were all patients examined for strength?

It is still not clear how MT and MS were classified as low, etc. Were these classifications based on other studies describing age and sex adjusted normal subjects? Also, what cut off values were used to indicate sarcopenia?  Please elaborate.

Table 2. The authors indicate in the manuscript regarding differences between subgroups; however, the table lacks post-hoc pair-wise comparisons between subgroups so we really don’t know which subgroups differ from each other. These analyses must be done to state any differences between subgroups.

Minor comments

Noradrenaline was misspelled numerous times in the manuscript.

The authors indicate that this was a retrospective study; however, measurement of muscle thickness via ultrasound and strength were conducted. Was this part of routine clinical practice? I don’t know of any centers that do this routinely which would negate the stated retrospective design.

I believe the authors failed to insert the true citation for reference 21 (propensity score matching) which then resulted in errors citing the wrong papers from reference 21 and greater. Please insert the correct reference and renumber references and citations throughout the manuscript.

Round 2

Reviewer 1 Report

The manuscript has been improved in the revised form, despite some limitations that now has been remarked and clarified  by the authors in the discussion. 

Reviewer 2 Report

The authors have adequately addressed my concerns.